# Transcriptome and Metabolome Analyses Revealed the Response Mechanism of Sugar Beet to Salt Stress of Different Durations

**DOI:** 10.3390/ijms23179599

**Published:** 2022-08-24

**Authors:** Jie Cui, Junliang Li, Cuihong Dai, Liping Li

**Affiliations:** 1School of Medicine and Health, Harbin Institute of Technology, Harbin 150086, China; 2College of Life and Environmental Science, Wenzhou University, Wenzhou 325035, China; 3College of Life Science, Northeast Forestry University, Harbin 150040, China

**Keywords:** RNA-seq, untargeted metabolomics, WGCNA, salt stress, sugar beet

## Abstract

Salinity is one of the most serious threats to agriculture worldwide. Sugar beet is an important sugar-yielding crop and has a certain tolerance to salt; however, the genome-wide dynamic response to salt stress remains largely unknown in sugar beet. In the present study, physiological and transcriptome analyses of sugar beet leaves and roots were compared under salt stress at five time points. The results showed that different salt stresses influenced phenotypic characteristics, leaf relative water content and root activity in sugar beet. The contents of chlorophyll, malondialdehyde (MDA), the activities of peroxidase (POD), superoxide dismutase (SOD), and catalase (CAT) were also affected by different salt stresses. Compared with control plants, there were 7391 and 8729 differentially expressed genes (DEGs) in leaves and roots under salt stress, respectively. A total of 41 hub genes related to salt stress were identified by weighted gene co-expression network analysis (WGCNA) from DEGs, and a transcriptional regulatory network based on these genes was constructed. The expression pattern of hub genes under salt stress was confirmed by qRT-PCR. In addition, the metabolite of sugar beet was compared under salt stress for 24 h. A total of 157 and 157 differentially accumulated metabolites (DAMs) were identified in leaves and roots, respectively. Kyoto Encyclopedia of Genes and Genomes (KEGG) pathway analysis further indicated that DEGs and DAMs act on the starch and sucrose metabolism, alpha-linolenic acid metabolism, phenylpropanoid biosynthesis and plant hormone signal transduction pathway. In this study, RNA-seq, WGCNA analysis and untargeted metabolomics were combined to investigate the transcriptional and metabolic changes of sugar beet during salt stress. The results provided new insights into the molecular mechanism of sugar beet response to salt stress, and also provided candidate genes for sugar beet improvement.

## 1. Introduction

Salinity is one of the most severe abiotic threats that affects the growth and development of crops [1,2]. Soil salinization is a growing problem in agriculture that can negatively alter the quality and yield of crops. The common effect of soil salinity on plants comes from the inhibition of growth by Na^+^ and Cl^−^ accumulation [3]. An excess of salt reduces the water potential on the root surface and prevents water absorption [4]. Some soils are actually hyperhaline for certain crop life; however, some salt tolerant varieties such as sugar beet (*Beta vulgaris* ssp. *vulgaris*) can adjust to these conditions and produce a good harvest. Sugar beet is one of the most important sugar crops in the world, and sugar from beets accounts for about 35% of the world’s sugar production [5]. The major agricultural sugar beet regions are concentrated in the northeast, northwest and north of China. There is an urgent need to further improve the salt tolerance of sugar beet crops to cope with the growing problem of soil salinization in these regions. As a recently domesticated crop, cultivated beet inherited certain salt-tolerance traits from its wild ancestor, *Beta vulgaris* ssp. *maritima* (referred to as *B. maritima* or ‘sea beet’) [6], and a moderate amount of salt can even promote the growth of sugar beets [7]. The breeding and propagation of salt-tolerant sugar beet is a promising agronomic and engineering solution to address these challenges.

Unlike other abiotic stresses, salinity causes both osmotic stress and ion toxicity in plants [8]. Plant growth may be rapidly impaired by osmotic stress in the first phase, and then specific ion toxicity, primarily from Na^+^ and Cl^−^ accumulation, may induce membrane disorganization, the generation of reactive oxygen species, metabolic toxicity, inhibition of photosynthesis, and the attenuation of nutrient acquisition in the second phase of salt stress [9,10]. In plants, excess ingestion of Na+ can alter the absorption of mineral nutrients, such as calcium (Ca^2+^), magnesium (Mg^2+^) and potassium (K^+^) [8,11,12], with adverse effects on enzyme catalytic activity and various metabolic pathways. Nevertheless, the growth and development of sugar beet is not significantly affected by absorbing large amounts of Na^+^. This is primarily due to the use of Na^+^ to replace the function of K^+^, such as long-distance transport of anions, stomatal regulation, and osmotic regulation in sugar beet [13,14,15]. In addition, sugar beet cells can accumulate large amounts of Na^+^ in the vacuole by ion compartmentalization [16].

Plants need to adjust their growth and development processes to cope with salt stress, and the reconstruction of transcriptions is a direct reflection of these reactions at the gene regulation level. Skorupa et al. found that genes related to translation and photosynthesis are upregulated by salinity in sugar beet leaves [17]. Using GC-MS, Hossain et al. also found that photorespiratory metabolism was stimulated in salt stressed sugar beet [18]. Lv et al. revealed the transcriptional changes in sugar beet monosomic addition line M14 under different salt stress concentrations (200 and 400 mM NaCl) [19]. Nevertheless, the reaction of plants to salt stress is a continuous process and existing studies lack continuous tracking of transcriptome changes in sugar beet under salt stress. The weighted gene co-expression network analysis (WGCNA) method, as a network approach, can group genes into specified modules based on the high correlations between co-expression genes across the samples, resulting in a cluster of genes that share a similar function. Several studies on the transcriptome data utilizing the WGCNA method to investigate complex traits in plants’ reactions to stress, such as salt stress [20], chilling stress [21], γ radiation stress [22] and biotic stress [23,24] have been reported.

Metabolomics is a new omics technology, developed after transcription and proteomics, which can qualitatively/quantitatively analyze changes in metabolite content in organisms [25]. Our previous study showed that the relative germination rate of cultivar ‘O68’ under salt treatment was more than 70% in 300 mmol·L^−1^ NaCl, whereas seedling growth in 200 mmol·L^−1^ NaCl was even stronger than in the control group [26]. With the objective of obtaining deeper insights into these resistance mechanisms in sugar beet, the present study employed RNA-seq utilizing cultivar O68 treated with 300 m mol·L^−1^ NaCl, and five treatment time points were selected to study the expression pattern of genes under salt stress. A comprehensive and integrated analysis of these different datasets has identified 7391 and 8729 differentially expressed genes (DEGs) in leaves and roots under salt stress, respectively. Additionally, untargeted metabolomics were used to detect the global changes of metabolites under 12 h of salt treatment in seedlings, ultimately identifying 157 and 157 differentially accumulated metabolites (DAMs) in leaves and roots, respectively. These data will improve understanding of the salt stress response of sugar beet.

## 2. Results

### 2.1. Salt Affects the Morphology and Physiological Indexes of Sugar Beet

In the present study, the treatment concentration was increased to 300 mmol·L^−1^ to provide adequate pressure, and exposure of seedlings to NaCl resulted in obvious morphological and physiological changes in both roots and leaves. With the prolongation of treatment time, leaves progressively displayed wilting and curling, and the aged leaves (the first pair of euphylla) were observed to turn yellow at 72 h (Appendix A). Analysis of the relative water content (RWC) indicated that RWC of leaves increased after 12 h, and then reduced continuously under salt stress (Figure 1a). The chlorophyll content was also raised at 12 h, 24 h and 48 h (Figure 1b).

Plant roots provide the ability to absorb, store and transport minerals and water, although roots can usually adjust to their environment, the accumulation of salt ions leads to cytotoxic effects [27]. A progressive color darkening of beet root was observed with increasing salt treatment time, and the root tip turned dark at 1 mm after 24 h treatment (Appendix A). In addition, a one-week rehydration experiment after 72 h salt treatment revealed that numbers of original roots were germinated from hypocotyls (Appendix A). Triphenyltetrazolium chloride (TTC) reduction was used to assess root activity in this study. The reduction activity of 24 h was 1.5 times greater than that of control plants (Figure 1c).

Under salt stress, the content of malondialdehyde (MDA) in leaves was raised at 12 h, then recovered at 24 h, and then increased again, while the content of MDA in roots significantly declined at 24 h, then recovered slightly at 48 h, and decreased again at 72 h (Figure 1d). Protective enzyme activity detection revealed that the activity of peroxidase (POD) was greatly strengthened under salt stress in both leaves and roots (Figure 1e). The activity of catalase (CAT) was significantly enhanced in leaves but reduced in roots (Figure 1f). The activity of superoxide dismutase (SOD) was diminished in leaves but did not change markedly in roots (Figure 1g).

### 2.2. Salt Affects the Ion Content of Sugar Beet

In order to evaluate the impact of salt stress on ion uptake in seedlings, ICP-MS was used to determine changes in the contents of four important elements (sodium, potassium, calcium and magnesium) in sugar beet. The results indicated that the level of sodium ions in leaves rose swiftly after 12 h of salt treatment, and then continued to increase after a slight recovery at 24 h (Figure 2a). In addition, potassium (Figure 2b) and calcium (Figure 2c) levels in leaves were significantly diminished after salt treatment, but no significant changes were observed for magnesium (Figure 2d). In the roots, salt treatment resulted in raised levels of sodium and calcium, and declining levels of potassium and magnesium (Figure 2).

### 2.3. Salt Affects the Gene Expression of Sugar Beet

Thirty transcriptome libraries were constructed from three-pairs-euphylla-stage seedlings treated with 300 mM NaCl for five different durations (0 h, 12 h, 24 h, 48 h and 72 h). Then, more than 1.35 billion reads were generated from RNA-seq, with approximately 45 million reads for each sample. After quality control, 97.6% (1.32 billion) of the reads comprising valid data were processed for further analysis. These reads were mapped on the sugar beet genome with an average mapping ratio of over 92%. The detailed information of RNA-seq data generated, valid data and mapped ratios are provided in Appendix A. After quantifying gene expression levels as fragments per kilobase of transcript per million mapped reads (FPKM), a total of 27,677 genes were identified as expressed genes in the present study. A violin plot was adopted to show the distribution of gene expression levels in each sample. The results showed that gene expression levels were generally comparable among different time point groups and the repeatability in-group was better (Appendix A). Correlation analysis showed that the correlation coefficient between repetitions within the group was greater than 0.9, the correlation coefficient between different salt treatment groups was between 0.8 and 0.9, and the correlation coefficient between salt stress treatment and control samples was between 0.6 and 0.8, implying that salt stress results in significant changes in the expression levels of a large number of genes in sugar beet (Appendix A).

Analysis of DEGs at different treatment durations illustrated that each salt treatment duration had its specific DEGs, and the time point with the largest number of specific DEGs was 12 h in both leaves and roots (Figure 3). In addition, 1057 and 2044 common DEGs were found at four treatment durations in leaves and roots, respectively, indicating that these genes may play a crucial role in the response to salt stress in sugar beet. Analysis of expression patterns indicated that most common DEGs were continuously upregulated or downregulated during salt stress (Appendix A). Gene ontology (GO) categories revealed that common DEGs were primarily enhanced in ‘regulation of transcription’, ‘transcription’, ‘oxidation-reduction process’, ‘defense response’ and ‘protein phosphorylation’ in both leaves and roots (Appendix A).

### 2.4. Co-Expression Network Analysis of Differentially Expressed Genes

After filtering out genes with low expression in all samples (FPKM < 1), 7391 and 8729 DEGs were screened from leaves and roots for co-expression network analysis, respectively. Co-expression modules were constructed by the expression values of DEGs in leaves and roots. On the whole, the sample hierarchical clustering plot in each sample was divided into two clusters using the flashClust tool package of the WGCNA algorithm method, and no outlier samples were obtained (Appendix A). A heat map view of the topological overlap measure (TOM) of co-expressed genes in different modules is shown in Appendix A. Finally, 11 co-expression modules were identified in the co-expression network in leaves and roots, respectively. Information about genes that belong to each module is listed in Appendix A. There was no gray module in leaves, while the gray module in roots contained 28 genes that could not belong to other modules, accounting for 0.32% of all DEGs in roots. 

Then, the correlation between modules and salt treatment intensity (duration) was calculated. Based on correlation and p-value, five (turquoise, brown, greenyellow, purple and blue) and four (turquoise, yellow, green and blue) most relevant modules were selected as critical modules for further analysis in leaves (Figure 4a) and roots (Figure 4b), respectively. Analysis of gene expression patterns showed that genes in the critical module were significantly differentially expressed at the time points with which the module was highly correlated (Appendix A). In addition, most genes in the turquoise module of leaves and roots were significantly downregulation under salt stress, while genes in other modules were significantly upregulated, compared to controls.

GO enrichment analysis indicated that there were significant differences in functional enrichment of genes between modules (Figure 5a,b). In the biological process category, DEGs of leaf module turquoise were significantly enriched in cytoplasmic translation (GO:0002181) and ribosomal large subunit assembly (GO:0000027), DEGs of leaf module brown were significantly enriched in negative regulation of ethylene-activated signaling pathway (GO:0010105) and ubiquitin-dependent protein catabolic process (GO:0006511), DEGs of leaf module blue were significantly enriched in proteolysis involved in cellular protein catabolic process (GO:0051603) and proteasome-mediated ubiquitin-dependent protein catabolic process (GO:0043161), DEGs of root module turquoise were significant enriched in oxalate metabolic process (GO:0033609) and cell surface receptor signaling pathway (GO:0007166), DEGs of root module green were significantly enriched in plant-type secondary cell wall biogenesis (GO:0009834) and fatty acid biosynthetic process (GO:0006633), DEGs of root module blue were significantly enriched in negative regulation of transcription, DNA-templated (GO:0045892). In the cellular component category, the term with the most DEGs in leaf module turquoise was chloroplast (GO:0009507), while nucleus (GO:0005634) was the term with the most DEGs in leaf module brown, leaf module blue and root module blue, and the modules with the most DEGs in root module turquoise were plasma membrane (GO:0005886) and integral component of membrane (GO:0016021). In the molecular function category, DEGs of leaf module turquoise were significantly enriched in the structural constituent of ribosome (GO:0003735), DEGs of leaf module brown were significantly enriched in carboxylyase activity (GO:0016831), and DEGs of leaf module blue were significantly enriched in cysteine-type endopeptidase activity. DEGs of root module turquoise were significantly enriched in oxalate decarboxylase activity (GO:0046564), and DEGs of root module blue were significantly enriched in nucleic acid binding (GO:0003676).

KEGG enrichment analysis demonstrated that DEGs in leaf module turquoise were substantial enriched in ribosome (ko03010), fatty acid biosynthesis (ko00061), porphyrin and chlorophyll metabolism (ko00860), biotin metabolism (ko00780), and lysine biosynthesis (ko00300) pathway. DEGs in leaf module brown were substantially enriched in glucosinolate biosynthesis (ko00966) and betalain biosynthesis (ko00965), and DEGs in leaf module blue were significantly enriched in other types of O-glycan biosynthesis (ko00514) and sulfur metabolism (ko00920) (Figure 5c). In roots, DEGs in module turquoise were considerably enhanced in sesquiterpenoid and triterpenoid biosynthesis (ko00909) and RNA polymerase (ko03020) (Figure 5d).

### 2.5. Hub Gene Identification and Expression Pattern Analysis

Hub genes, a few highly interconnected genes in a co-expression module, were considered to be biologically significant [28]. In the present study, the connectivity value of the genes in in each critical module and CytoHubba analysis were used to evaluate hub genes related to salt tolerance. Finally, a total of 41 hub genes were identified for all critical modules (Table 1). It is interesting to note that some hub genes belong to transcription factors (TFs), such as bHLH35, ERF054. The transcriptional level salt stress response network was constructed using hub genes and DEGs in leaves (Figure 6a) and roots (Figure 6b), respectively. STRING database was used to annotate the interactions among the encoded proteins of these genes.

### 2.6. Validation of DEGs by qRT-PCR

To validate the RNA-seq data, twenty hub genes were randomly selected to detect their expression levels by qRT-PCR from the same batch of RNA samples for sequencing. Most of the determination coefficients (R^2^ value) of the qRT-PCR validation data against the FPKM data were significantly positively correlated, and 70% of the tested hub genes had a. R^2^ value greater than 0.9 (Figure 7). These results indicate the high reliability of high-throughput sequencing.

### 2.7. Salt Affects the Metabolomic of Sugar Beet

Considering the time point of maximum physiological and transcriptional difference, the 12 h treatment condition (300 mmol·L^−1^ NaCl for 12 h) was selected in this study to investigate the metabolites changes under salt stress. Untargeted metabolomics were carried out in BL_st/BL_ck and BR_st/BR_ck. For leaves, principal component analysis (PCA) between treatment group and control group displayed 47.87% and 49.34% total variation (PC1+PC2) in leaves and roots, implying a significant difference in metabolites between the st and ck group (Appendix A). No over-fitting was obtained by the replacement test (200 times) for PLS-DA models (Appendix A), suggesting that the identification of DAMs is highly reliable. For leaves, 157 DAMs belonging to 12 super classes were identified with 108 upregulated and 49 downregulated metabolites (Figure 8a, Appendix A). For roots, 157 DAMs belonging to 11 super classes were identified with 112 upregulated and 45 downregulated metabolites (Figure 8b and Appendix A). KEGG analysis indicated that DAMs in leaves were significantly enhanced in linoleic acid metabolism (ko00591) (Appendix A), while DAMs in roots were significantly enriched in plant hormone signal transduction (ko04075) (Appendix A).

### 2.8. Profiles of DEGs and DAMs under Salt Stress in Sugar Beet

The joint KEGG enrichment analysis between transcriptome (12 h/ck) and metabolome revealed 5 and 8 comapped significant enrichment pathways in leaves (Figure 9a) and roots (Figure 9b), respectively. Spearman’s correlation coefficients between the DEGs and DAMs were estimated, and the top 60 of the correlation coefficients were further selected and described as a heat map (Figure 9c,d).

## 3. Discussion

Salt stress is the main abiotic stress that reduces crop yield, while sugar beet, as a salt-tolerant cash crop, can cope with this situation. Therefore, it is important to explain the molecular mechanisms underlying the salt reaction in beet. High-throughput sequencing and combination analysis of multi-omics provide an effective solution to analyze the molecular mechanism of stress response in plants. In the present study, a total of 11,992 DEGs (Figure 3) and 295 DAMs (Figure 8) were obtained in response to salt stress by RNA-seq and untargeted metabolomics. Another new finding of this investigation is the construction of time- and tissue-related modules in sugar beet under salt stress. WGCNAs serve as gene modules for locating synergistic expressions and can analyze the relationship between gene expression and stress intensity (duration). Five and four key modules corresponding to the salt treatment time in leaves and roots were gained. In the gene network, we found that some crucial receptor kinases, calcium signaling, transcription factors, redox enzymes and heat shock proteins appear in the hub genes.

For plants, exposure to salt stress normally results in a lag phase of growth, which is followed by a growth recovery phase after adjusting to stress [29]. The relative water content of the leaves was significantly reduced under salt stress, implying that the seedlings were under osmotic stress. Plants mostly synthesize and accumulate osmoprotectants to coordinate osmotic balance. In the present study, a 2-fold increase in betaine was found in leaves after 12 hours of salt stress (Appendix A). Meanwhile, *choline monooxygenase* (*Bv6_146100_wdro*) and *betaine-aldehyde dehydrogenase* (*Bv5_116230_ntjn* and *Bv7_178050_meus*) involved in betaine synthesis [30] were considerably upregulated under salt stress. These results suggest that betaine may play an important role in the osmotic regulation of leaves.

POD, SOD and CAT are essential protective enzymes in plants. Under salt stress, the activity of POD enzyme increased more than 5-fold in roots (Figure 1e), which was consistent with the significant upregulation of hub gene *peroxidase 66* (*Bv3_066080_qmgn*). Photosynthesis will produce a large amount of ROS (reactive oxygen species), which is catalyzed by SOD to generate H_2_O_2_ and then catalyzed by CAT to convert into water. This study found that CAT activity was significantly strengthened in leaves under salt stress, which was consistent with the upregulation of catalase gene (*Bv1_021080_ggjt*) expression in leaves. Unlike in leaves, CAT enzyme activity in roots was significantly reduced after salt stress, which was consistent with the downregulated expression of CAT encoding gene (*Bv1_021020_qtds*) in roots (Figure 1f). These results suggest that CAT and POD may play an important role in removing ROS in leaves and roots, respectively. In addition, the transcripts and metabolomics revealed that genes involved in phenylpropanoids and flavonols pathways were activated, leading to the accumulation of multiple phenolic compounds, which may have the potential for ROS scavenging [31].

Intracellular Ca^2+^ is the most important second messenger. When a plant is in contact with the initial reaction of NaCl, the stimulation of too much salt in the environment can be converted in a few seconds into a change of the Ca^2+^ concentration in cytoplasm [32,33]. The Ca^2+^ signal is then transmitted and decoded by calcium-dependent protein kinases (CDPK), calmodulin (CaM)/CaM-like proteins, calcineurin B-like proteins (CBLs), and annexins to trigger downstream cellular responses [34]. Various protein kinases play a central role in plant growth, development and stress response, and are widely involved in a variety of complex signal transduction pathways [35,36]. It was found that the calcium content in roots increased markedly after salt stress (Figure 2c). Transcriptome studies showed that the expression levels of more than 300 genes encoding various protein kinases were significantly altered under salt stress, such as *CDPK 26* (*Bv9_215750_yqnm*), *calmodulin-like protein 2* (*Bv2_035110_kyjk*), *CBL-interacting serine/threonine-protein kinase 14* (*Bv7_157470_urmk*) and *annexin-like protein RJ4* (*Bv2_041310_rnso*), etc. Additionally, the expressions of one hub gene *calmodulin-like protein 1* (*Bv_008200_zqtr*) in roots were significantly upregulated under salt stress. This gene may play an important role in calcium signal transduction in sugar beets under salt stress.

Production of reactive oxygen species (ROS) and abscisic acid (ABA), and the activation of the salt overly sensitive (SOS) pathway for Na^+^ exclusion and ion homeostasis control at the cellular level, are also stimulated by raised cytosolic Ca^2+^ [29,37]. Metabolomic results indicated that the ABA content of sugar beet leaves and roots was markedly increased under salt stress. The rapid accumulation of ABA in cells depended on the cleavage of carotenoid precursors by dioxygenase [38], and transcriptomic results revealed that two *9-cis-epoxycarotenoid dioxygenase genes NCED1* (*Bv_01350_texx*) and *NCED5* (*Bv_004050_xzzo*), were significantly upregulated in sugar beet (Figure 6). In addition, *respiratory burst oxidase homolog protein A* (*Bv5_099740_zpio*), as a hub gene in leaves, and *respiratory burst oxidase homolog protein B* (*Bv4_075730_oajz*) in roots were also significantly upregulated during salt stress. These genes may play an important role in salt stress-induced ABA accumulation in sugar beet.

The accumulation of ABA leads to the activation of downstream effectors, including TFs and ion channels, to implement important adaptive responses such as stomatal closure, osmoprotectant synthesis, and the induction of various stress-responsive genes. Salt-responsive genes are commonly categorized into ABA-independent or ABA-dependent groups. ABA-dependent genes generally contain a conserved ABA-responsive cis-acting element (ABRE) in their promoter regions. ABREs can be recognized by transcription factors containing a basic leucine zipper structure (bZIP). Our RNA-seq data revealed that a number of bZIP coding genes, such, as *Bv3_050990_kqjn*, *Bv9_214630_fnpa*, *Bv1_013750_smoy*, etc. were highly expressed under salt stress. ABA-independent genes generally contain dehydration responsive elements/C-repeats (DRE/CRTs) in their promoters and are regulated by AP2-EREBP transcription factors. Two hub genes, *AP2-EREBP transcription factors Bv6_135390_gwrr* and *Bv9_208590_puum*, were markedly upregulated in roots under salt stress. In addition, two *dehydration-responsive element-binding proteins, DREB1* (*Bv3_066590_ignp*) and *DREB2* (*Bv2_044600_drzk*), belonging to the AP2-EREBP family, were also significantly upregulated under salt stress in sugar beet. It is proposed that they may play an important role in regulating the expression of downstream salt tolerance genes.

Propagation of the ROS signal depends on activation of respiratory burst oxidase homolog, which is controlled by ethylene, ABA-dependent protein kinase and phosphatidic acid (PA) binding [39]. Transcriptome revealed that hub gene *non-specific phospholipase C1* (*NPC1*) was considerably upregulated under salt stress in leaves (Table 1), while *NPC2* (*Bv4_073710_zzht*) was significantly upregulated in roots. Recent studies have shown that *NPCs* conferred lipid-mediated signaling during the salt stress response [40]. NPC can catalyze the hydrolysis of phosphatidylcholine (PC) and phosphatidylethanolamine (PE) to generate diacylglycerol (DAG) [41]. Metabolomics results indicated that the relative contents of PC and PE compounds were considerably reduced under salt stress (Figure 9), while their hydrolysates LysoPC and LysoPE were significantly increased. DAG can be further catalyzed by diacylglycerol kinase (DGK) to generate PA, and the expression of *DGK1* (*Bv5_113810_fdmf*) and *DGK2* (*Bv4_078480_dtwj*) were considerably raised during salt stress. These results indicate that salt stress activates NPC to catalyze the hydrolysis of PC and PE to generate large amounts of DAG, which in turn can be catalyzed by DGK to produce PA. DAG and PA can further activate the expression and activation of downstream genes and proteins. In general, Ca^2+^, DAG and PA play an important role as signal molecules in the signal transduction process of sugar beet under salt stress.

KEGG analysis of numerous DEGs and DAMs led to membrane regulation. Choline is an important component of biofilms. The first step of choline synthesis is to generate ethanolamine from serine catalyzed by serine decarboxylase (SDC) [42]. Subsequently, ethanolamine is catalyzed by ethanolamine kinase (CEK) to generate phosphoethanolamine [43]. Ultimately, choline synthesis is catalyzed by phosphoethanolamine N-methyltransferase (PEAMT) [44]. Under salt stress, sugar beet promoted the synthesis of choline by upregulating the expression of *SDC* (*Bv1_011320_dejg*), *CEK* (*Bv8_194710_kpip*) and *PEAMT* (*Bv5_106830_tcqr*, *Bv5_111890_duxs* and *Bv8_184180_kkck*) genes (Appendix A), and then regulated the membrane structure and synthesis of betaine. In addition, transcriptomic and metabolomic results showed that *linoleic acid 13S-lipoxygenase 2-1* (*Bv4_072070_uwja*) was upregulated and catalyzed the synthesis of 13-oxoode linoleic acid from linoleic acid, which may play an important role in the regulation of plasma membrane fluidity.

## 4. Materials and Methods

### 4.1. Plant Growth and Treatment

The seeds were obtained from our own laboratory (Heilongjiang, China). They were immersed in water for ten hours, then sterilized in 0.1% (*v/v*) HgCl_2_ for 10 min, washed repeatedly with distilled water, and germinated on damp filter paper in a germination box at 26 °C for 2 days. After germination, seedlings were transferred to plastic pots (43.5 cm × 20 cm × 14 cm, 10plants per pot) filled with half-strength Hoagland solution. The germinating seeds were cultivated under a 16/8 light photoperiod at 24 °C (day)/18 °C (night) in a phytotron (Friocell 707, MMM, Brno, CZ). We have previously demonstrated that the cultivar ‘O68’ can normally grow under 300 mM NaCl [26]. In order to explore the genome-wide dynamic response to salt stress in sugar beet, three-pair-euphylla-stage seedlings were treated with 300 mmol·L^−1^ NaCl. The treatment methods are shown in Appendix A, to exclude the influence caused by different growth times during sampling as much as possible. All collected samples were immediately frozen in liquid nitrogen for half an hour and then stored at −80 °C until further use.

### 4.2. Physiologic Indexes Detection and Ion Content Detection

Fresh leaves from the third pair of euphylla were used to detect relative water content by (W_fresh_ − W_dry_)/(W_saturation_ − W_dry_) × 100%. Chlorophyll was extracted with acetone and measured by UV-2100 (Unico, Shanghai, China), referring to the Naeem method [45]. The content of malondialdehyde (#G0109W, Grace Biotechnology Co., Ltd, Suzhou, China), POD activity (#G0107W, Grace Biotechnology Co., Ltd, Suzhou, China), SOD activity (#G0104W, Grace Biotechnology Co., Ltd, Suzhou, China) and CAT activity (#G0106W, Grace Biotechnology Co., Ltd, Suzhou, China) were detected according to the standard protocol. Fresh roots (0.1 g) were collected for the determination of root activity by the TTC reduction method [46]. Data were obtained using BioTek Epoch (BioTek, Highland Park, IL, USA) and each treatment was repeated three times. The ion content of digested samples was determined using ICP MASS SPECTROMETES (Thermo Fisher Scientific, Bremen, Germany). All experiments were conducted in triplicate for each treatment. Each treatment sample was taken from a mixed pool consisting of six plants.

### 4.3. RNA Extraction and Transcriptome Sequencing

To reduce the error caused by individual differences as much as possible, each sample used to construct the library was taken from six plants. Total RNA was extracted from each sample using TRIZOL (Invitrogen, Carlsbad, CA, USA), according to the manufacturer’s instructions. The quantity and purity of total RNA were analyzed using a Bioanalyzer 2100 and RNA 6000 Nano LabChip Kit (Agilent, Palo Alto, CA, USA) with RIN number >7.0. Approximately 1 μg of high quality total RNA from each sample was subjected to isolate Poly (A) mRNA with Poly-T oligo attached magnetic beads (Invitrogen, Carlsbad, CA, USA). Following purification, the mRNA was fragmented into small pieces by utilizing divalent cations under elevated temperatures. Then, the cleaved RNA fragments were reverse-transcribed to generate the ultimate cDNA library, in accordance with the protocol for the mRNA Seq sample preparation kit (Illumina, San Diego, CA, USA). The average insert size for the paired-end libraries was 300 bp (±50 bp). A total of 30 libraries (leaf_ck, leaf_12, leaf_24, leaf_48, leaf_72, root_ck, root_12, root_24, root_48, root_72) were constructed and then paired-end sequencing was performed using Illumina Novaseq™ 6000 by LC Sciences (Hangzhou, China), following the vendor’s recommended protocol. All of the sequencing data were deposited in the NCBI Short Read Archive (SRA) database under the BioProject ID: PRJNA666117.

### 4.4. Analysis of Sequencing Data

The raw reads obtained from sequencing were processed to remove primer/adaptor contamination, low quality bases and undetermined bases using Cutadapt [47]. Then, the sequence quality was verified with FastQC (http://www.bioinformatics.babraham.ac.uk/projects/fastqc/) (accessed on 8 December 2019). The clean reads were mapped to reference genome RefBeet-1.2 (http://bvseq.boku.ac.at/Genome/Download/RefBeet-1.2/) (accessed on 9 December 2019), using hierarchical indexing for spliced alignment of transcripts (HISAT) [48]. The mapped reads of each sample were assembled using StringTie [49]. After the definitive transcriptome was generated, StringTie was used to estimate the expression levels of all transcripts by calculating FPKM (fragments per kilobase of transcript per million mapped reads). The differential expression analysis was performed utilizing edgeR R-packages [50]. DEGs were screened by one-way ANOVA analysis among five groups. All transcripts were annotated by conducting BLASTx similarity searches against the NCBI non-redundant protein database (Nr), with an e-value threshold of 10^−5^. UpSetR was used to visualize differences in gene expression patterns between groups [51].

### 4.5. Construction of Gene Co-Expression Network Analysis

In this study, DEGs with FPKM >1 were screened for gene co-expression network construction and co-expression modules identification using the WGCNAR-package (http://www.r-project.org/) (accessed on 22 March 2020). The process follows the procedures as described in a previous study [52]. A standard scale-free network was established by the condition of scale-free topology index > 0.9. The soft threshold power in this study was set as 9 and 17 for leaves and roots, respectively. Subsequently, the WGCNA algorithm was used to construct the co-expression modules with the one-step network construction method and extract the gene information in each module. Ultimately, the co-expression module structure was visualized using the WGCNA plot Dendrogram and Colors function. Connectivity between genes in each module was calculated. The module sample associations were estimated using the correlation between modules and the duration of salt treatment. The GO [53] and KEGG [54] databases were used to annotate the biological and functional properties of DEGs in each module.

### 4.6. Establishment of Candidate Hub Genes and Construction of the Gene Networks

In general, genes with the highest connectivity (top 5%) were designated as hub genes. Herein, protein–protein interaction networks were also used to assess hub genes, as reported in Nature [55]. The protein–protein interaction (PPI) relationships between DEGs were analyzed using the STRING database with a confidence score > 0.4 set as the cut-off threshold [56]. Based on PPI analysis results, the Cytoscape plug-in CytoHubba was further used to evaluate hub genes for each module by a total of 12 algorithms, and the genes recognized by more than half algorithms (≥6) were selected as candidate hub genes. Eventually, hub genes for each module were screened according to connectivity and CytoHubba statistics, and the interaction network was visualized using Cytoscape ver. 3.6.9 [57].

### 4.7. Real-Time Quantitative Reverse Transcription PCR (RT-qPCR) Analysis

For validation of the gene expression gained from high-throughput sequencing, qRT-PCR was performed on randomly selected genes. All primers are listed in Appendix A. The qRT-PCR reactions were performed using High-Capacity cDNA Reverse Transcription Kits (Thermo Fisher SCIENTIFIC, Foster City, CA, USA) and iTaq Universal SYBR^®^ Green Supermix (BIO-RAD, Hercules, CA, USA) on the CFX Real-time PCR system (BIO-RAD, Singapore, SG). According to our previous study [58], *PP2A*+ *UBQ5* and *PP2A*+*25S RNA* genes were used as endogenous controls for root and leaf, respectively. To avoid non-specific amplification, a melting curve was carried out for each PCR product. The expression level of the genes in different samples was calculated by the comparative 2^−^^△△CT^ method.

### 4.8. Metabolite Analysis

Metabolites were extracted on ice by 50% methanol buffer. Then, the mixture of metabolites was vortexed for 1 min and incubated at room temperature for 10 min. The mixture was centrifugated at 4000× *g* for 20 min and the supernatant was recycled for the LC-MS analysis. Quality control (QC) samples were prepared by combining 10 μl of each extraction mixture. All samples were analyzed using a TripleTOF 5600 Plus high resolution tandem mass spectrometer (SCIEX, Warrington, UK) with both positive and negative ion modes by LC Sciences (Hangzhou, China) following their standard procedures [59]. The acquired LC-MS data was analyzed by XCMS, using R software [60]. The KEGG, in-house (http://spldatabase.saskatoonlibrary.ca/) (accessed on 2 February 2021) and HMDB (http://www.hmdb.ca/) (accessed on 2 February 2021) databases were used to perform level-one and level-two identification and annotation for metabolites. MetaX (http://metax.genomics.cn/) (accessed on 3 February 2021) was used for QC processing of peak intensity data of ion signal extracted from XCMS. A *t*-test was used to calculate the p-value, which was then corrected by FDR (Benjamini–Hochberg). In addition, partial least squares discrimination analysis (PLS-DA) was used to determine the relationship between metabolite expression level and sample type. Variable importance for the Projection (VIP) was calculated to assess the influence of each metabolite expression model on the classification and discrimination of each group of samples, so as to assist the screening of DAMs. Metabolites with |log2FC| ≥ 1, *q* ≤ 0.05, and VIP ≥ 1 were defined as DAMs.

### 4.9. Statistical Analysis

The physiological experiments, RNA-seq and qRT-PCR were conducted in triplicate (*n* = 3), and statistical analysis of the data from the control and different plant treatments was performed by analysis of variance (one-way ANOVA) or independent-samples *t*-test using SPSS software (18.0) or edgeR. A probability value of *p* < 0.05 was considered to denote a statistically significant difference. Data are presented as the mean ± standard deviation (SD) of three replicates. Metabolomics experiments were conducted in sextuplicate (*n* = 6).

## 5. Conclusions

The physiology, transcriptome and metabolite analysis of sugar beet leaves and roots under different salt stress conditions were compared. Different salt stresses influenced phenotypic characteristics, the contents of MDA, the activity of antioxidant enzymes, gene expression and the accumulation of metabolites in beet seedlings. A total of 41 hub genes were identified by WGCNA in response to salt stress. Sugar beet transmitted stress signals by regulating Ca^2+^, DAG and PA, thus regulating the salt stress response process. The alpha-linolenic acid metabolism pathway plays an important role in the process of sugar beet salt stress response. Future studies could further investigate the influence of the hub gene on the growth, development and stress tolerance of sugar beet. These results provide a broader and better understanding of the metabolic processes underlying different salt stress responses and the hub genes identified in this research provide potential targets for future molecular breeding.

## Figures and Tables

**Figure 1 ijms-23-09599-f001:**
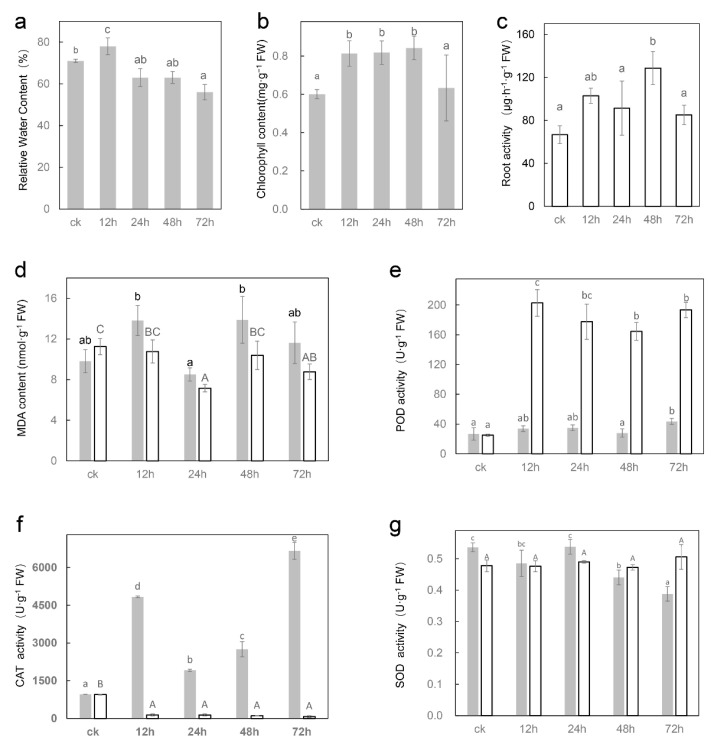
The physiological indexes of sugar beet O68 seedlings under different durations of salt treatments. (**a**) relative water content (**b**) chlorophyll content (**c**) root activity (**d**) MDA content (**e**) POD activity (**f**) CAT activity (**g**) SOD activity. The gray bars represent leaves and the white bars represent roots. The significance of leaves is marked with lowercase letters and that of roots is marked with uppercase letters. Triplicate biological replicates were performed. Different letters indicate significant differences according to Student−Newman−Keuls (*p* < 0.05).

**Figure 2 ijms-23-09599-f002:**
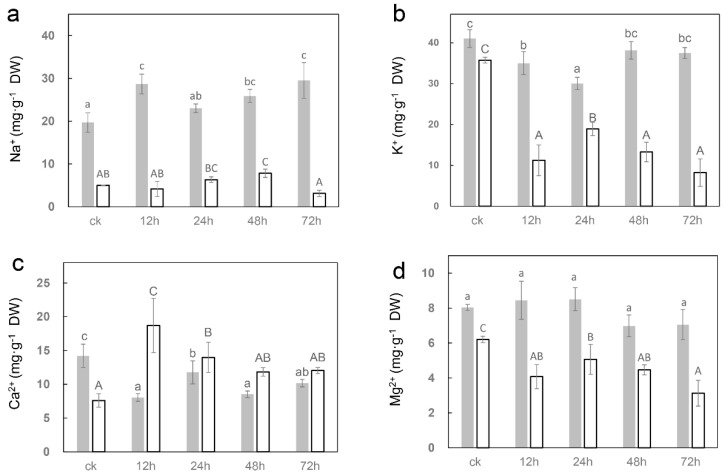
The changes in ion content under salt treatments in sugar beet O68 seedlings. (**a**) sodium (**b**) potassium (**c**) calcium (**d**) magnesium. The gray bars represent leaves and the white bars represent roots. The significance of leaves is marked with lowercase letters and that of roots is marked with uppercase letters. Triplicate biological replicates were performed. Different letters indicate significant differences according to Student−Newman−Keuls (*p* < 0.05).

**Figure 3 ijms-23-09599-f003:**
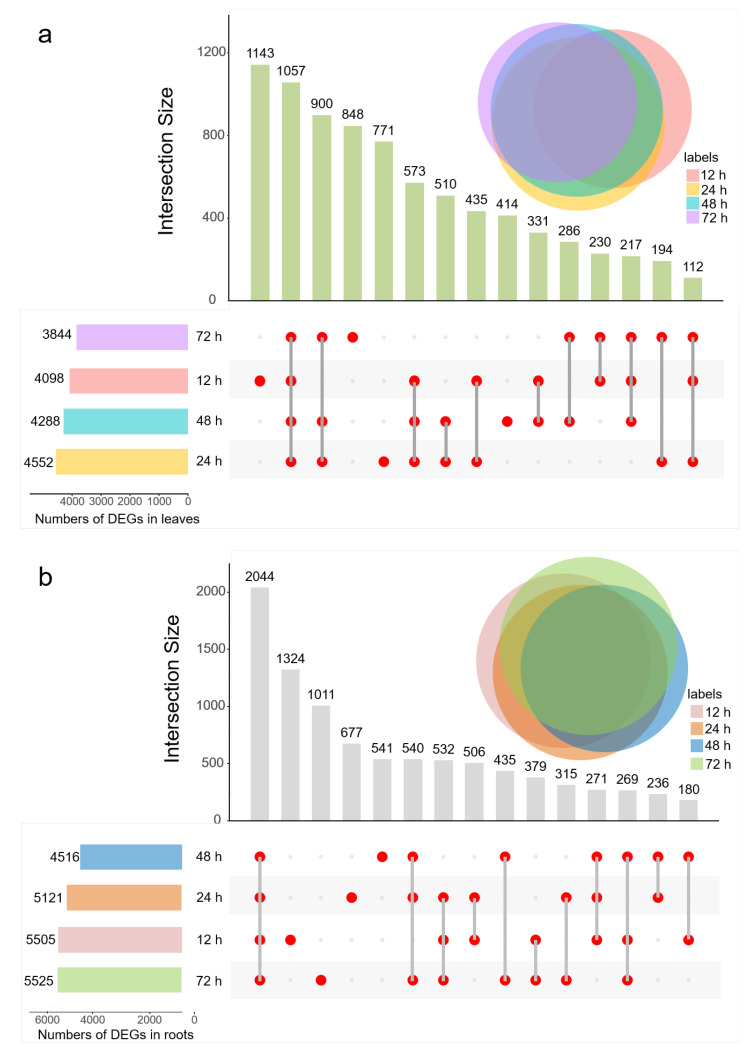
Distribution of DEGs among samples treated with salt for different durations. (**a**) leaves (**b**) roots.

**Figure 4 ijms-23-09599-f004:**
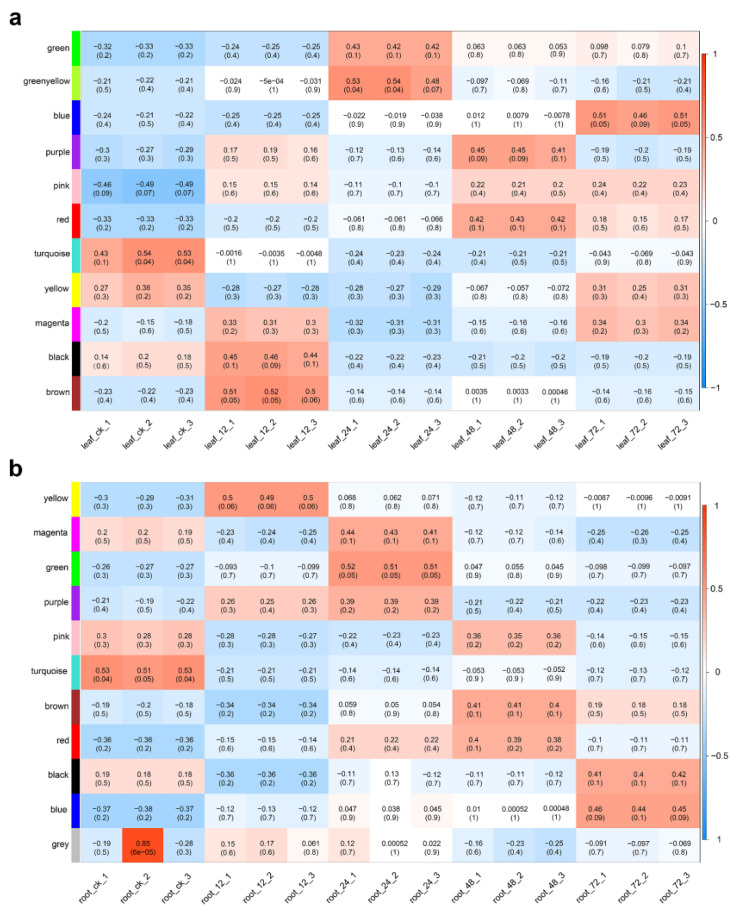
Correlation between modules and each sample in co-expression networks in leaves (**a**) and roots (**b**). Each row represents a module eigengene, and each column a sample. Each cell contains the corresponding correlation and *p*-value.

**Figure 5 ijms-23-09599-f005:**
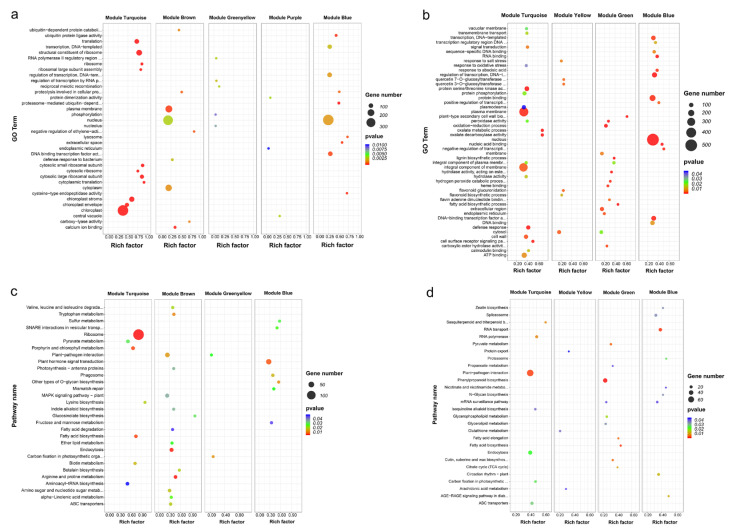
Functional enrichment analysis for genes in each critical module.GO enrichment for critical modules in leaves (**a**) and roots (**b**); KEGG enrichment for critical modules in leaves (**c**) and roots (**d**).

**Figure 6 ijms-23-09599-f006:**
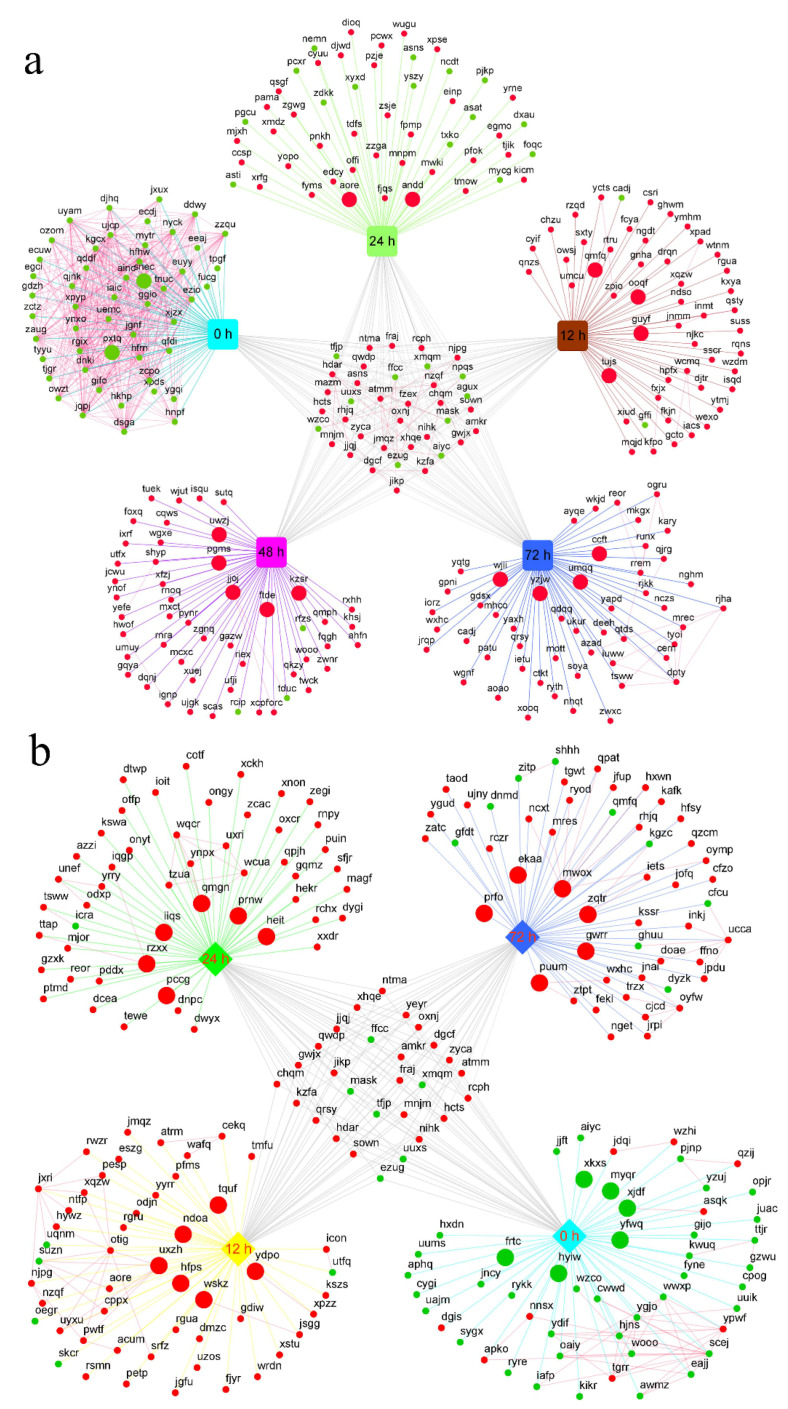
Transcriptional network of leaves (**a**) and roots (**b**) in responses to salt stress.The square represents the module corresponding to each color; Large dots represent hub genes, small dots represent DEGs; Red represents upregulation, green represents downregulation; The red line represents the interaction relationship annotated by the STRING database, and the other colored lines represent the affiliation relationship between genes and corresponding modules.

**Figure 7 ijms-23-09599-f007:**
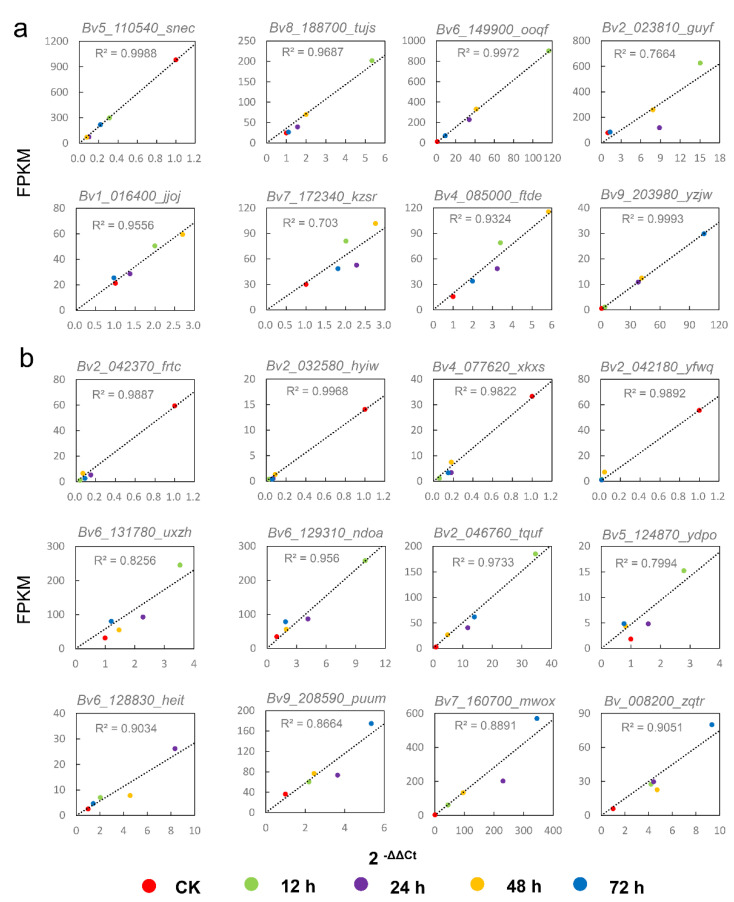
Comparison between qRT-PCR and deep sequencing of hub genes under salt stress. (**a**) leaf. (**b**) root. Triplicate biological replicates were performed. The *X*-axis represents qRT-PCR results (relative quantity calculated by 2^−ΔΔCT^), and the *Y*−axis represents RNA−SEQ results (FPKM).

**Figure 8 ijms-23-09599-f008:**
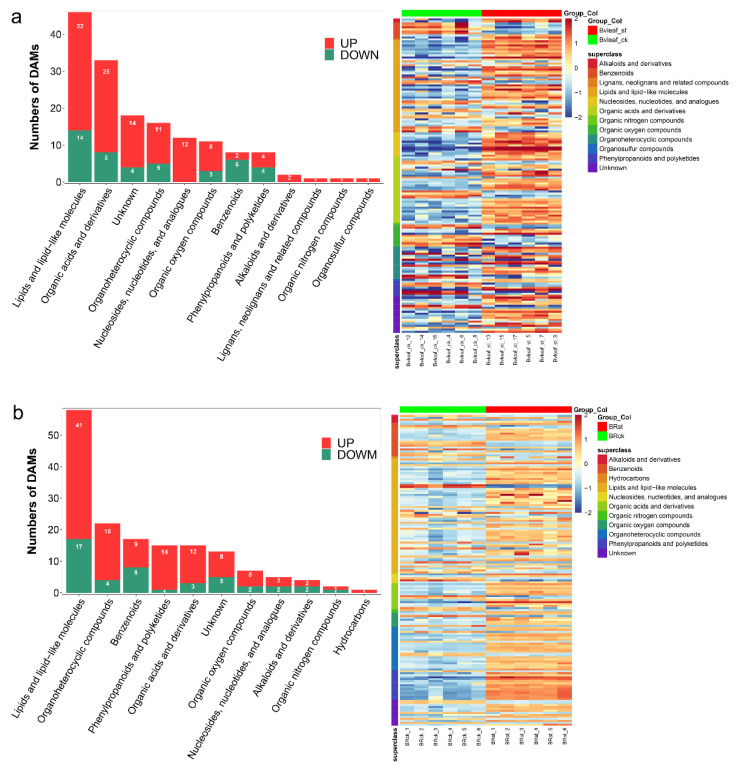
Differentially accumulated metabolites under salt treatment in sugar beet. (**a**) leaf. (**b**) root. Six independent replicates of each stage are also displayed in the heat map.

**Figure 9 ijms-23-09599-f009:**
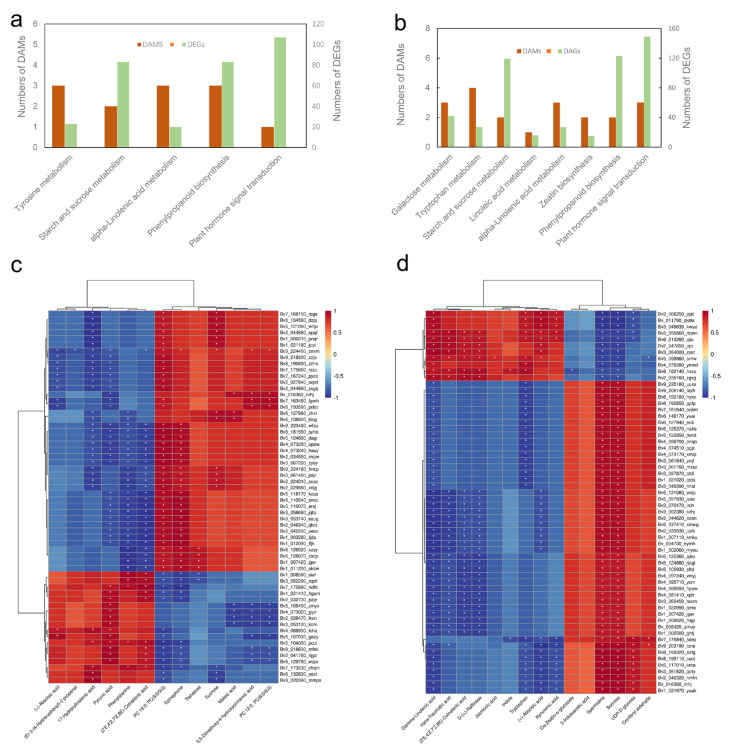
Overview of the joint analysis of DEGs and DAMs. The joint KEGG enrichment analysis between DEGs and DAMs in leaves (**a**) and roots (**b**). Correlation analysis of DEGs and DAMs in the above pathways in leaves (**c**) and roots (**d**). Asterisks represent *p* ≤ 0.05.

**Table 1 ijms-23-09599-t001:** The hub genes detected in WGCNA modules in leaves and roots.

Module	Gene ID	Annotation
Leaves		
Turquoise	Bv2_029170_pxtq	cycloartenol-C-24-methyltransferase
	Bv5_110540_snec	beta-glucosidase 13
Brown	Bv8_188700_tujs	miraculin
	Bv6_149930_qmfq	shikimate kinase, chloroplastic
	Bv2_023810_guyf	DExH-box ATP-dependent RNA helicase DExH17
	Bv6_149900_ooqf	GEM-like protein 5
Greenyellow	Bv4_074970_andd	thioredoxin F-type, chloroplastic
	Bv4_095830_aore	probable glutathione S-transferase
Purple	Bv1_016400_jjoj	chaperone protein dnaJ 11, chloroplastic
	Bv4_086990_pgms	probable ubiquitin-conjugating enzyme E2 26
	Bv_007910_uwzj	transcription factor bHLH35
	Bv7_172340_kzsr	non-specific phospholipase C1
	Bv4_085000_ftde	B-box zinc finger protein 22
Blue	Bv9_203980_yzjw	two-pore potassium channel 3
	Bv2_033120_umqq	vacuolar-processing enzyme
	Bv2_042060_ccft	probable choline kinase 2
	Bv6_139660_wjii	homeobox-leucine zipper protein ATHB-40
Roots		
Turquoise	Bv4_090510_myqr	L-type lectin-domain containing receptor kinase S.7
	Bv2_042370_frtc	LRR receptor-like serine/threonine-protein kinase
	Bv_016430_xjdf	heavy metal-associated isoprenylated plant protein 47
	Bv2_032580_hyiw	ankyrin repeat-containing protein NPR4
	Bv4_077620_xkxs	7-deoxyloganetic acid glucosyltransferase-like
	Bv2_042180_yfwq	cytosolic sulfotransferase 5
Yellow	Bv1_009510_hfps	WD repeat-containing protein C2A9.03-like
	Bv6_131780_uxzh	glutamate decarboxylase 4
	Bv6_129310_ndoa	AAA-ATPase At3g50940
	Bv3_067190_wskz	peroxisomal fatty acid beta-oxidation multifunctional protein MFP2
	Bv2_046760_tquf	vacuolar amino acid transporter 1
	Bv5_124870_ydpo	protein DETOXIFICATION 48
Green	Bv2_044780_rzxx	3-ketoacyl-CoA synthase 10
	Bv1_016480_prnw	protein IN2-1 homolog B
	Bv4_083640_iiqs	slit homolog 3 protein
	Bv2_043550_pccg	ABC transporter G family member 32
	Bv6_128830_heit	epoxide hydrolase 4
	Bv3_066080_qmgn	peroxidase 66
Blue	Bv4_091130_ekaa	hypothetical protein BVRB_4g091130
	Bv6_135390_gwrr	ethylene-responsive transcription factor RAP2-4
	Bv9_208590_puum	ethylene-responsive transcription factor ERF054
	Bv7_160700_mwox	AF4/FMR2 family member 4
	Bv6_133000_prfo	uncharacterized protein LOC104895442
	Bv_008200_zqtr	calmodulin-like protein 1

## Data Availability

All sequencing data were deposited in the NCBI Short Read Archive (SRA) database under the BioProject ID: PRJNA666117. Relevant supporting data can be found within the article and additional files.

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
