# Peer review of "Transcriptome and Metabolome Analyses Revealed the Response Mechanism of Sugar Beet to Salt Stress of Different Durations"

_ijms, 2022, doi:10.3390/ijms23179599_

Round 1
Reviewer 1 Report
In the current study, Transcriptome and metabolome analyses revealed the response mechanism of sugar beet to salt stress. The study topic is very interesting, novel, and helpful for the readerships of IJMS. The overall manuscript is well organized and well-written. However, the main concerns about this manuscript can be found below, and minor revision are suggested.
In abstract section full form of abbreviations must be written at first use.
Mainly words are repeated in a single sentence like “and” must be revised.
In introduction section add mechanism, effects and losses due to salinity stress.
Line 55 must be cited. The following references could help the authors.
https://doi.org/10.3390/ijms22179175, https://doi.org/10.1007/s10725-021-00785-7,
Discuss economic and industrial importance of sugar beet.
Also discuss losses of sugar beet yield due to salinity stress.
Also discuss growth conditions and origin of sugar beet.
Result and discussion are well presented. The authors are advised to avoid long sentences.
Author Response
In abstract section full form of abbreviations must be written at first use.
Response: Weighted Gene Co-Expression Network Analysis (WGCNA) has been added to abstract.
Mainly words are repeated in a single sentence like “and” must be revised.
Response: The excess ‘and’ has been replaced with ‘,’. The corrected sentence is as follows: The contents of chlorophyll, malondialdehyde (MDA), the activities of peroxidase (POD), superoxide dismutase (SOD), and catalase (CAT) were also affected by different salt stresses.
In introduction section add mechanism, effects and losses due to salinity stress.
Response: Thank you for your advice. ‘An excess of salt reduces the water potential on the root surface and prevents water absorption [4].’ has been added to line 37. ‘In plants, excess ingestion of Na+ can alter the absorption of mineral nutrients, such as calcium (Ca2+), magnesium (Mg2+) and potassium (K+) [8, 11, 12], with adverse effects on enzyme catalytic activity and various metabolic pathways.’ has been added to line 57.
Line 55 must be cited. The following references could help the authors.
https://doi.org/10.3390/ijms22179175, https://doi.org/10.1007/s10725-021-00785-7,
Response: The references has been cited as required in the article.
Discuss economic and industrial importance of sugar beet.
Also discuss losses of sugar beet yield due to salinity stress.
Also discuss growth conditions and origin of sugar beet.
Response: ​Thank you for your advice on writing. ‘Sugar beet is one of the most important sugar crops in the world, sugar from beets ac-counts for about 35% of the world's sugar production [5]. The major agricultural re-gions of sugar beet were concentrated in the northeast, northwest and north of China. There is an urgent need to further improve the salt tolerance of sugar beet crops to cope with the growing problem of soil salinization in these regions.’ has been added to line 41.
Reviewer 2 Report
The presented manuscript "Transcriptome and metabolome analyzes revealed the response mechanism of sugar beet to salt stress" by Jie Cui , Junliang Li , Cuihong Dai and Liping Li makes it possible to evaluate data on the likely effect of salinity on gene expression. The inability to detect a tissue-specific pattern and data on the level of ions in tissues makes this work somewhat preliminary. However, a detailed analysis of the genetic response was performed qualitatively and can be taken into account in the further work of these or other authors.
Unfortunately, without taking into account the structural organization of individual layers of plant cells and tissues, these data cannot be adequately assessed.
It is desirable that an attempt be made to describe the effects of salinity, even though the physiological response to salt exposure includes significant cumulative changes and usually differs after 10 or more days of exposure. In this work, we can only talk about the evaluation of primary reactions. This is probably due to a typical logical error characteristic of molecular biological work, in which no distinction is made between sensitivity to stress and resistance, so sensitivity does not mean the absence of resistance, and non-sensitivity can be characteristic of both resistant and unstable forms.
In this case, the species under study can potentially be resistant, which can be associated with both a specific model of resistance and a non-specific one. Unfortunately, the article ignores the difficulty of identifying these types of resistance, although the data themselves make it possible to isolate the genes associated with the formation of the oxidative, toxic, and osmotic effects characteristic of salinity.
It seems to me that the title of the article does not correspond to its content, since it is not obvious that we are dealing with stress and it is not obvious how this time range was chosen. In addition, there are no photographs and descriptions of the physiological effect of "stress" in the work. It is possible that the authors should expand the section on methods and explain why this concentration and duration was chosen. Why was only one concentration used? It might be worth confirming this choice with a link. In addition, there is a feeling that the data obtained relate to one experiment, it is also not clear how many plants were used and how many repetitions were carried out. Without photographs and clarification of these data, the result looks somewhat preliminary.
In the introduction, it is not clear how exactly this work can help in understanding, if it is not obvious what kind of result should be achieved. The authors want plants not to die, or sugar beets not to lose biomass under salt stress, or sugar losses to be reduced. The way in which the results (even preliminary ones) are applied is not reflected in the discussion. Meanwhile, the details are important because they are not self-evident.
These remarks require a significant correction of the introduction, methodological part and discussion. After making changes to the title and clarifications to the manuscript, it can be accepted.
Author Response
It seems to me that the title of the article does not correspond to its content, since it is not obvious that we are dealing with stress and it is not obvious how this time range was chosen.
Response: The title has been corrected as ‘Transcriptome and metabolome analyses revealed the response mechanism of sugar beet to salt stress of different duration’.
In addition, there are no photographs and descriptions of the physiological effect of "stress" in the work.
Response: Thank you for your advice. We have provided high-resolution images of the leaves and roots of the sugar beet in Fig. S1.
It is possible that the authors should expand the section on methods and explain why this concentration and duration was chosen. Why was only one concentration used? It might be worth confirming this choice with a link.
Response: Thank you for your advice. ‘We have previously demonstrated that the cultivar 'O68' can normally grow under 300mM NaCl [43]. In order to explore the genome-wide dynamic response to salt stress in sugar beet, three-pair-euphylla-stage seedlings were treated with 300 mmol·L-1 NaCl. The treatment methods are shown in Table S6 to exclude the influence caused by different growth times during sampling as much as possible.’ has been added to line 412.
In addition, there is a feeling that the data obtained relate to one experiment, it is also not clear how many plants were used and how many repetitions were carried out. Without photographs and clarification of these data, the result looks somewhat preliminary.
Response: Thank you for your advice. ‘All these experiments were conducted in triplicate for each treatment. Each treatment sample was taken from a mixed pool consisting of six plants.’ has been added to line 430. ‘Triplicate biological replicates were performed. Different letters indicate significant differences according to Student-Newman-Keuls (P < 0.05).’ has been added to line 124 and line 139. ‘Triplicate biological replicates were performed.’ has been added to line 260.
In the introduction, it is not clear how exactly this work can help in understanding, if it is not obvious what kind of result should be achieved. The authors want plants not to die, or sugar beets not to lose biomass under salt stress, or sugar losses to be reduced.
Response: Thank you for your advice. ‘Our previous study showed that the relative germination rate of cultivar ‘O68’ under salt treatment was more than 70% in 300 mmol L-1 NaCl, whereas seedling growth in 200 mmol·L-1 NaCl was even stronger than in the control group [26].With an objective to get deeper insights into these resistance mechanisms in sugar beet, the present study employed RNA-seq utilizing cultivar ‘O68’ treated with 300 m mol·L-1 NaCl, and five time-points of treatment were selected to study the expression pattern of genes under salt stress. ’ has been added to line 81.
Reviewer 3 Report
This article presented transcriptomic and metabolomics analysis to reveal sugar beet responses against salinity stress. The study is well organized and data is well arranged. The findings would be helpful for future studies. Before recommending this article for publication, there are some shortcomings for that should be resolve.
General comments
Overall, the study is well designed and presented in a good way, but mostly the literature is not cited. Grammatical and typos must be revised
Abstract
Methods are not well presented in the abstract. Which transcriptomic and metabolomics characterizations were analyzed.
Also add quantitative results in this section.
What is “WGCNA”?
What is the benefit of this study? As many studies have already presented the same techniques.
Introduction
The introduction part is well written but still some details are required. The authors should provide details of the beeta vulgaris,
Its economic importance.
Threats of salinity stress.
Habitat and stress response mechanisms.
Impact of NaCl on sugar beet.
The importance of transcriptomic and metabolomics studies on sugar beet and other plants.
Advance techniques of transcriptomic and metabolomics.
Materials and methods
Section 2.4. could be cited with
https://doi.org/10.1007/s10725-021-00785-7, 10.1016/j.micpath.2020.103966,
section 4.4 and 4.5 could be cited with the following article.
https://doi.org/10.3390/ijms22179175,
Results
Results are well presented.
What does significant mean should be added in the legends? At which value like (P<0.05) SD was determined
Discussion and conclusion
Discussion and conclusion are well presented. However, future recommendations based on the obtained results must be added in the conclusion section.
Author Response
Abstract
Methods are not well presented in the abstract. Which transcriptomic and metabolomics characterizations were analyzed.
Response: Thank you for your advice. We have corrected two sentences as required. ‘In the present study, physiological and transcriptome analyses of sugar beet leaves and roots were compared under salt stress at 5 time points.’ has been added to line 12. ‘In addition, the metabolite of sugar beet was compared under salt stress for 24 hours. A total of 157 and 157 differentially accumulated metabolites (DAMs) were identified in leaves and roots, respectively.’ has been added to line 21.
Also add quantitative results in this section.
Response: Thank you for your advice. ‘The expression pattern of hub genes under salt stress was confirmed by qRT-PCR.’ has been added to line 21.
What is “WGCNA”?
Response: Thank you for your advice. Weighted Gene Co-Expression Network Analysis (WGCNA) has been added to line 19.
What is the benefit of this study? As many studies have already presented the same techniques.
Response: Thank you for your advice. ‘In this study, RNA-seq, WGCNA analysis and untargeted metabolomics were combined to investigate the transcriptional and metabolic changes of sugar beet during salt stress. The results pro-vided new insights into the molecular mechanism of sugar beet response to salt stress, and also provided candidate genes for sugar beet improvement.’ has been added to line 26.
Introduction
The introduction part is well written but still some details are required. The authors should provide details of the beeta vulgaris,
Its economic importance.
Response: Thank you for your advice. ‘Sugar beet is one of the most important sugar crops in the world, sugar from beets ac-counts for about 35% of the world's sugar production [5]’ has been added to line 41.
Threats of salinity stress.
Response: Thank you for your advice. ‘There is an urgent need to further improve the salt tolerance of sugar beet crops to cope with the growing problem of soil salinization.’ has been added to line 44.
Habitat and stress response mechanisms.
Response: Thank you for your advice. ‘In addition, sugar beet cells can accumulate large amounts of Na+ in the vacuole by ion compartmentalization [16].’ has been added to line 62.
Impact of NaCl on sugar beet. NaCl
Response: Thank you for your advice. ‘As a recently domesticated crop, cultivated beets inherited certain salt-tolerance traits from its wild ancestor Beta vulgaris ssp. maritima (referred to as B. maritima or ‘sea beet’) [6], a moderate amount of salt can even promote the growth of sugar beets [7].’ has been added to line 45.
The importance of transcriptomic and metabolomics studies on sugar beet and other plants.
Response: Thank you for your advice. ​‘Skorupa et al. found that genes related to translation and photosynthesis are up-regulated by salinity in sugar beet leaves [17]. Using GC-MS, Hossain et al. also found that photorespiratory metabolism was stimulated in salt-stressed sugar beet [18]. Lv et al. revealed the transcriptional changes in sugar beet monosomic addition line M14 under different salt stress concentrations (200 and 400 mM NaCl) [19].’ has been added to line 66.
Advance techniques of transcriptomic and metabolomics.
Response: Thank you for your advice. ​‘Metabolomics is a new omics technology developed after transcription and proteomics, which can qualitatively / quantitatively analyze changes in metabolite content in organisms [25]. ’ has been added to line 79.
Materials and methods
Section 2.4. could be cited with
https://doi.org/10.1007/s10725-021-00785-7, 10.1016/j.micpath.2020.103966,
Response: Thank you for your advice. References have been added as requested.
section 4.4 and 4.5 could be cited with the following article.
https://doi.org/10.3390/ijms22179175,
Response: Thank you for your advice. We regret that we have not found a connection between this reference and sections 4.4 and 4.5. ​Please confirm whether the link is correct or give me more detailed guidance.
Results
Results are well presented.
What does significant mean should be added in the legends? At which value like (P<0.05) SD was determined
Response: Thank you for your advice. ‘Triplicate biological replicates were performed. Different letters indicate significant differences according to Student-Newman-Keuls (P < 0.05).’ has been added to line 124 and line 139. ‘Triplicate biological replicates were performed.’ has been added to line 260.
Discussion and conclusion
Discussion and conclusion are well presented. However, future recommendations based on the obtained results must be added in the conclusion section.
Response: Thank you for your advice. ‘Future studies can further investigate the influence of the hub gene on the growth, development and stress tolerance of sugar beet.’ has been added to line 535.